# F-OAL: Forward-only Online Analytic Learning with Fast Training and Low Memory Footprint in Class Incremental Learning

**Huiping Zhuang**[1]*, **Yuchen Liu**[2]*, **Run He**[1], **Kai Tong**[1], **Ziqian Zeng**[1],
**Cen Chen**[1,4]†, **Yi Wang**[3], **Lap-Pui Chau**[3]

[1]South China University of Technology, China
[2]The University of Hong Kong, Hong Kong SAR
[3]The Hong Kong Polytechnic University, Hong Kong SAR
[4]Pazhou Lab, Guangzhou, China
{hpzhuang,runhe,kaitong,zqzeng,cenchen}@scut.edu.cn,
liuyuchen@connect.hku.hk,
{yi-eie.wang, lap-pui.chau}@polyu.edu.hk

## Abstract

Online Class Incremental Learning (OCIL) aims to train models incrementally, where data arrive in mini-batches, and previous data are not accessible. A major challenge in OCIL is Catastrophic Forgetting, i.e., the loss of previously learned knowledge. Among existing baselines, replay-based methods show competitive results but requires extra memory for storing exemplars, while exemplar-free (i.e., data need not be stored for replay in production) methods are resource-friendly but often lack accuracy. In this paper, we propose an exemplar-free approach—Forward-only Online Analytic Learning (F-OAL). Unlike traditional methods, F-OAL does not rely on back-propagation and is forward-only, significantly reducing memory usage and computational time. Cooperating with a pre-trained frozen encoder with Feature Fusion, F-OAL only needs to update a linear classifier by recursive least square. This approach simultaneously achieves high accuracy and low resource consumption. Extensive experiments on benchmark datasets demonstrate F-OAL's robust performance in OCIL scenarios. Code is available at: https://github.com/liuyuchen-cz/F-OAL

## 1  Introduction

Class Incremental Learning (CIL) updates the model incrementally in a task-by-task manner with new classes in the new task. Traditional CIL most plans for static offline datasets which historical data are accessible. However, with the rapid increase of social media and mobile devices, massive amount of image data have been produced online in a streaming fashion, and render training models on static data less applicable. To address this, Online Class Incremental Learning (OCIL) is developed taking an online constraint in addition to the existing CIL setting. OCIL is a more challenging CIL setting in which data come in mini-batches, and the model is trained only in one epoch (i.e., learning from one pass data stream) [14]. The model is required to achieve high accuracy, fast training time, and low resource consumption [25].

However, CIL techniques (including OCIL) suffer from Catastrophic Forgetting (CF) [27], also known as the erosion of previous knowledge when new data are introduced. The problem becomes

---

*Equal contribution.
†Corresponding author.

38th Conference on Neural Information Processing Systems (NeurIPS 2024).

more severe in online scenarios since the model can only see data once. Two major factors contribute to CF: (1) Using the loss function to update the whole network leads to uncompleted feature capturing and diminished global representation [11]. (2) Using back-propagation to adjust linear classifier results in *recency bias*, which is a significantly imbalanced weight distribution, showing preference only on current learning data [15].

To address CF in an online setting, replay-based methods [9, 21] are the mainstream solution by preserving old exemplars and revisiting them in new tasks. This strategy has strong performance but is resource consuming, while exemplar-free methods [17, 20] have lower resource consumption but show less competitive results.

Recently, Analytic Continual Learning (ACL) [49] methods emerged as an exemplar-free branch, delivering encouraging outcomes. ACL methods pinpoint the iterative back-propagation as the main factor behind catastrophic forgetting and seek to address it through linear recursive strategies. Remarkably, for the first time, these methods achieve outcomes comparable to those utilizing replay-based techniques.

There are two limitations in existing ACL methods: (1) Multiple iterations of base training are needed when the model is applied. Subsequently, the acquired knowledge is encoded into a *regularized feature autocorrelation matrix* by analytic re-alignment. The incremental learning phase then unfolds, utilizing the recursive least squares method for updates. This pattern is repeated when the dataset is switched, significantly elevating the temporal cost in an online scenario. (2) Classic ACL methods demand data aggregation from a single task, facilitating analytic learning in one fell swoop. This process increases GPU memory footprint and is unsuitable for online contexts where data for each task is presented as mini-batches.

To address those limitations, we propose Forward-only Online Analytic Learning (F-OAL) that learns online batch-wise data streams. The F-OAL consists of a frozen pre-trained encoder and an Analytic Classifier (AC). The frozen encoder is capable of countering the uncompleted feature representation caused by using the loss function to update and replace the time-consuming base training. With Feature Fusion and Smooth Projection, the encoder provide informative representation for analytic learning. The AC is updated by recursive least square rather than back-propagation to solve recency bias and decrease calculation. Therefore, F-OAL is an exemplar-free countermeasure to CF and reduces resource consumption since the encoder is frozen and only the AC is updated.

Our main contributions can be concluded as follows:

• We present the F-OAL, an exemplar-free technique that achieves high accuracy and low resource consumption together for the OCIL.

• F-OAL redefines the OCIL problem into a recursive least square manner and is updated in a mini-batch manner.

• F-OAL introduces a framework of frozen pre-trained encoder with Feature Fusion to generate representative features and Smooth Projection for recursively updating AC to counter CF.

• We conduct massive experiments on benchmark datasets with other OCIL baselines. The results demonstrate that F-OAL achieves competitive results with fasting training speed and low GPU footprint.

## 2 Related Works

**Online Class Incremental Learning** focuses on extracting knowledge from one pass data stream with new classes in the new task. Time and memory consumption requirements are particularly critical in OCIL, given the fast and large nature of online data streams [13].

**Replay-based methods** [33, 25, 21, 9, 31, 12, 2, 11, 15, 41, 37, 36, 1, 38, 19, 6, 39, 35, 9] are mainstream solutions for OCIL problems by preserving historical exemplars and using them in new tasks. The accuracy is better than exemplar-free methods', but the training time and memory consumption are higher.

**Analytic Learning** (AL), also referred to as pseudo-inverse learning [10], emerges as a solution to the pitfalls of back-propagation by recursive least square. Analytic Learning's computational intensity is demanding since the entire dataset is processed. The obstacle is solved by the block-wise recursive

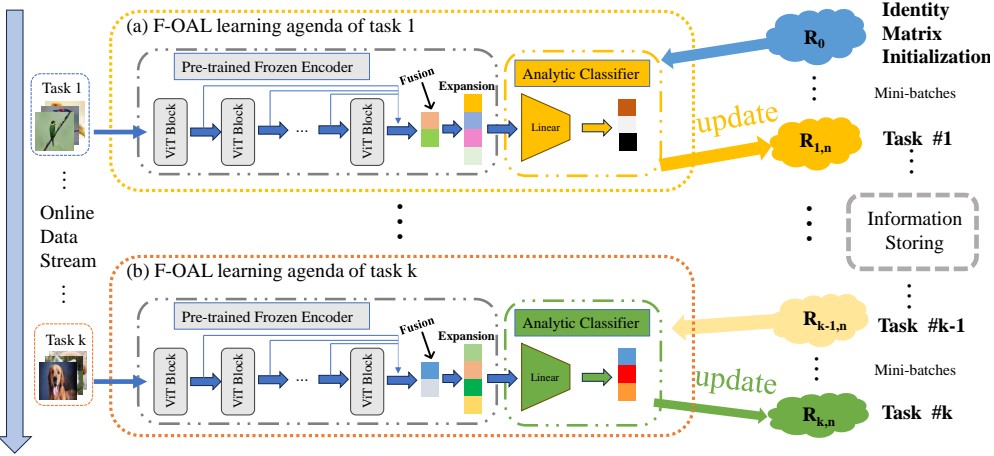

Figure 1: This diagram illustrates the learning agenda of F-OAL. In the encoder , features from each block of the ViT are extracted, summed, and averaged to form a composite feature. This feature is then randomly projected into a higher-dimensional space and normalized using the sigmoid function, serving as the activation for updating the classifier. All parameters in the encoder remain frozen. In the analytic classifier section, we introduce $R$ to retain historical information and update the linear classifier using recursive least squares. This process is forward-only with no gradients.

Moore-Penrose algorithm [47], which achieves equivalent precision with joint-learning (i.e., training with all data). AL has already been used in online reinforcement learning [22], which shows the potential of AL in OCIL.

**Exemplar-free methods** can be categorized into regularization-based, prototype-based, and recently emerged ACL methods. **Regularization-based methods** [20, 17, 42, 8, 44] apply constraints on the loss function or change the gradients to preserve knowledge. These solutions reduce resource consumption but commonly can not outperform replay-based methods. **Prototype-based methods** [43, 45, 29, 40, 32, 7, 28] mitigate CF by augmenting the prototypes of features to classify old classes or generating psudo-features of old classes from new representations. **ACL methods** represent a new branch of the CIL community and show great potential in the OCIL scenario. Analytic Class Incremental Learning (ACIL) [49] is the first approach that applies AL to the CIL problem. The ACIL achieves a competitive performance in the offline CIL scenario. Gaussian Kernel Embedded Analytic Learning (GKEAL) [48], following the ACIL, focuses on solving CF in the few-shot scenario by adopting the kernel method. DS-AL [46] overcomes the under-fitting problem of AL.

## 3 Methodology

### 3.1 Proposed Framework

Our F-OAL framework consists of two modules:

**Encoder**. Encoder has two parts: backbone with Feature Fusion and Smooth Projection. The performance of analytic learning is highly dependent on the quality of the extracted features. Therefore, we employ Vision Transformer (ViT) [5] as the backbone instead of CNN, because ViT generates more comprehensive feature representation [30]. To further enhance the representativeness of the features, we utilize Feature Fusion. Specifically, we extract the outputs from each block of the ViT, sum them, and take their average to form the image feature.

Subsequently, we expand this feature into a higher-dimensional space using random projection and apply the sigmoid function for smoothing (i.e., Smooth Projection). This results in the final activation

used to update the classifier. Therefore, the encoder $\phi(\cdot)$ is defined as:

$$\phi(x) = \sigma \left( LP \left( \overbrace{\frac{B_1(x) + B_1(B_2(x)) + \cdots + B_n(B_{n-1}(\cdots B_1(x)))}{n}}^{\text{Feature Fusion}} \right) \right), \tag{1}$$

$$\underbrace{\phantom{\phi(x) = \sigma \left( LP \left( \frac{B_1(x) + B_1(B_2(x)) + \cdots + B_n(B_{n-1}(\cdots B_1(x)))}{n} \right) \right)}}_{\text{Smooth Expansion}}$$

where $B_1(\cdot) \cdots B_n(\cdot)$ are blocks in ViT, $LP(\cdot)$ is linear projection and $\sigma(\cdot)$ is sigmoid activation function.

Backbone is pre-trained and frozen and weight matrix of projection is sampled from normal distribution and is frozen. Freezing the encoder avoids selective learning caused by loss function, which prioritizes features that are easiest to learn rather than the most representative [11] and significantly reduces the number of trainable parameters.

**Analytic Classifier**. Unlike back-propagation, we employ the least squares method to obtain the analytic solution for the linear mapping from the activation to the one-hot label. We then recursively update the weight matrix of the linear mapping. This approach is forward-only and does not require the use of gradients, resulting in low memory usage and fast computation speed.

## 3.2 Analytic Solution

| Name | Description | Dimension |
|---|---|---|
| $\phi(X)$ | Activation of all images. | $V \times D$ |
| $Y$ | One-hot label of all images. | $V \times M$ |
| $\hat{W}$ | Joint-learning result of classifier's weight matrix. | $M \times D$ |
| $X_{k,i}^{(a)}$ | Activation matrix of the i-th batch of the k-th task. | $S \times D$ |
| $Y_{k,i}^{\text{train}}$ | One-hot label matrix of the i-th batch of the k-th task. | $S \times C_s$ |
| $X_{k,1:i}^{(a)}$ | Activation matrix from the start to the i-th batch of the k-th task. | $V_s \times D$ |
| $Y_{k,1:i}^{\text{train}}$ | One-hot label matrix from the start.to the i-th batch of the k-th task. | $V_s \times C_s$ |
| $\hat{W}^{(k,i)}$ | Classifier of the i-th batch of the k-th task. | $C_s \times D$ |
| $R_{k,i}$ | Regularized feature autocorrelation matrix to the n-th batch of the k-th task. | $D \times D$ |

Table 1: Description of notations and their dimensions. Here, $V$ is the number of all images, $D$ is the encoder output dimension, $M$ is the number of all classes, $S$ is the batch size, $C_s$ is the number of classes seen so far, and $V_s$ is the number of images seen so far.

The overview is shown in Figure 1. To derive our solution, the notations needed are listed in Table 1. Unlike other methods that treat model training as a back-propagation process, our method formulates the problem using linear regression, which allows for direct computation of the optimal parameters in a closed-form solution:

$$\boldsymbol{Y} = \phi(\boldsymbol{X})\boldsymbol{W}, \tag{2}$$

where $\phi(\cdot)$ is the pre-trained and forzen encoder and $\boldsymbol{W}$ is the learnable parameter of linear classifier. The learning problem can be rewritten into an optimization form:

$$\underset{\boldsymbol{W}}{\arg\min} \ \|\boldsymbol{Y} - \phi(\boldsymbol{X})\boldsymbol{W}\|_{\text{F}}^2 + \gamma \|\boldsymbol{W}\|_{\text{F}}^2, \tag{3}$$

where $\|\cdot\|_{\text{F}}$ represents the Frobenius norm and $\gamma$ is the regularization term. The optimal solution is defined as:

$$\hat{\boldsymbol{W}} = \left(\phi(\boldsymbol{X})^{\text{T}}\phi(\boldsymbol{X}) + \gamma\boldsymbol{I}\right)^{-1}\phi(\boldsymbol{X})^{\text{T}}\boldsymbol{Y}. \tag{4}$$

### 3.3 Learning Agenda

Given dataset $\{X_k^{\text{train}}, Y_k^{\text{train}}\}_1^m$ be the training set of task $k$ ($k = 1, 2, \dots, m$). Every task is divided into $n$ mini-batches. We denote $\{X_{k,i}^{\text{train}}, Y_{k,i}^{\text{train}}\}$, as the $i$ mini-batch ($i=1, 2, \dots, n$) of training set of task $k$.

At task 1 batch $k$, the model aims to optimize

$$\operatorname*{argmin}_{W^{(1,i)}} \left\| Y_{1,1:i}^{\text{train}} - (X_{1,1:i}^{(a)}) W^{(1,i)} \right\|_{\text{F}}^2 + \gamma \left\| W^{(1,i)} \right\|_{\text{F}}^2, \tag{5}$$

where

$$X_{1,1:i}^{(a)} = \phi(X_{1,1:i}^{\text{train}}). \tag{6}$$

The optimal solution to parameter $W^{(1,i)}$ is given as:

$$\hat{W}^{(1,i)} = \left( (X_{1,1:i}^{(a)})^{\text{T}} X_{1,1:i}^{(a)} + \gamma I \right)^{-1} (X_{1,1:i}^{(a)})^{\text{T}} Y_{1,1:i}^{\text{train}}. \tag{7}$$

Regarding $X_{1,1:i}^{(a)}$ and $Y_{1,1:i}^{\text{train}}$ as stacks of row activation vectors and one-hot label vectors respectively, we observe that $(X_{1,1:i}^{(a)})^{\text{T}}(X_{1,1:i}^{(a)})$ and $(X_{1,1:i}^{(a)})^{\text{T}} Y_{1,1:i}^{\text{train}}$ are both sums of outer products w.r.t feature vectors. Notice that the solution contains no data correlation. Thus we can further derive Equation 7 into

$$\hat{W}^{1,i} = \left( \left[ (X_{1,1:i-1}^{(a)})^{\text{T}} \quad (X_{1,i}^{(a)})^{\text{T}} \right] \begin{bmatrix} X_{1,1:i-1}^{(a)} \\ X_{1,i}^{(a)} \end{bmatrix} + \gamma I \right)^{-1} \left[ (X_{1,1:i-1}^{(a)})^{\text{T}} \quad (X_{1,i}^{(a)})^{\text{T}} \right] \begin{bmatrix} Y_{1,1:i-1}^{\text{train}} & \mathbf{0} \\ \mathbf{0} & Y_{1,i}^{\text{train}} \end{bmatrix} \tag{8}$$

$$= \left( (X_{1,1:i-1}^{(a)})^{\text{T}}(X_{1,1:i-1}^{(a)}) + (X_{1,i}^{(a)})^{\text{T}}(X_{1,i}^{(a)}) + \gamma I \right)^{-1} \left[ (X_{1,1:i-1}^{(a)})^{\text{T}} Y_{1,1:i-1}^{\text{train}} + (X_{1,i}^{(a)})^{\text{T}} Y_{1,i}^{\text{train}} \right].$$

Let

$$R_{1,i-1} = \left[ (X_{1,1:i-1}^{(a)})^{\text{T}}(X_{1,1:i-1}^{(a)}) + \gamma I \right]^{-1}. \tag{9}$$

be the *regularized feature autocorrelation matrix* at batch *i*-1 of task 1, where both historical and current information is encoded in.

Therefore, we can calculate both $R$ and $W$ in a recursive manner by the following theorems when the serial numbers of tasks and batches are given:

**Theorem 3.1** For the batch 1 of task $k$, Let $\hat{W}^{(0)}$ be the null matrix. Let $\hat{W}^{(k-1,n)'} = \left[ \hat{W}^{(k-1,n)} \quad \mathbf{0} \right]$ and $\hat{W}^{(k,1)}$ can be calculated via:

$$\hat{W}^{(k,1)} = \hat{W}^{(k-1,n)'} + R_{k,1} X_{k,1}^{(a)\text{T}} \left( Y_{k,1}^{\text{train}} - X_{k,1}^{(a)} \hat{W}^{(k-1,n)'} \right), \tag{10}$$

where

$$R_{k,1} = R_{k-1,n} - R_{k-1,n} X_{k,1}^{(a)\text{T}} \left( I + X_{k,1}^{(a)} R_{k-1,n} X_{k,1}^{(a)\text{T}} \right)^{-1} X_{k,1}^{(a)} R_{k-1,n}. \tag{11}$$

**Theorem 3.2** For the batch $i$ ( $i > 1$ ) of task $k$, $\hat{W}^{(k,i)}$ can be calculated via:

$$\hat{W}^{(k,i)} = \hat{W}^{(k,i-1)} + R_{k,i} X_{k,i}^{(a)\text{T}} \left( Y_{k,i}^{\text{train}} - X_{k,i}^{(a)} \hat{W}^{(k,i-1)} \right), \tag{12}$$

where

$$R_{k,i} = R_{k,i-1} - R_{k,i-1} X_{k,i}^{(a)\text{T}} \left( I + X_{k,i}^{(a)} R_{k,i-1} X_{k,i}^{(a)\text{T}} \right)^{-1} X_{k,i}^{(a)} R_{k,i-1}. \tag{13}$$

*Proof.* See Appendix A. $\qquad\square$

Thus, we achieve absolute memorization in an exemplar-free way with all data used only once. The learning agenda of F-OAL is summarised in Algorithm 1.

**Algorithm 1** Forward-Only Analytic Learning

---

**Input:** Mini-batches $\{X_{k,i}^{\text{train}}, Y_{k,i}^{\text{train}}\}_{1,1}^{m,n}$.

**Initialization:** Identity matrix $R_0$; Null matrix $W^{(0)}$.

**for** $k = 1$ **to** $m$ **do**
    **for** $i = 1$ **to** $n$ **do**
        **if** $i = 1$ **then**
            # **Theorem 3.1**:
            Load $\hat{W}^{(k-1,n)}$ and $R_{k-1,n}$;
            Expand $\hat{W}^{(k-1,n)}$ to $\hat{W}^{(k-1,n)'}$;
            Obtain activation $X_{k,1}^{(a)}$ based on $X_{k,1}^{\text{train}}$;
            Update $R_{k,1}$ by $X_{k,1}^{(a)}$ and $R_{k-1,n}$;
            Update $W^{(k,1)}$ by $X_{k,1}^{(a)}$; $Y_{k,1}^{\text{train}}$, $W^{(k-1,n)'}$ and $R_k$;
        **else**
            # **Theorem 3.2**:
            Load $W^{(k,i-1)}$ and $R_{k,i-1}$;
            Obtain activation $X_{k,i}^{(a)}$ based on $X_{k,i}^{\text{train}}$;
            Update $R_{k,i}$ by $X_{k,i}^{(a)}$ and $R_{k,i-1}$;
            Update $W^{(k,i)}$ by $X_{k,i}^{(a)}$; $Y_{k,i}^{\text{train}}$, $W^{(k,i-1)}$ and $R_{k,i-1}$;
        **end if**
    **end for**
**end for**

---

### 3.4 Complexity Analysis

In terms of space complexity, our trainable parameters are only $R$ and $W$. The $R$ matrix is of size $D \times D$, where $D$ is the output dimension of the encoder. In our paper, the encoder output dimension is 1000. Therefore, according to Equation 9, the size of the $R$ matrix is $1,000 \times 1,000$. The $W$ matrix has dimensions of $C \times D$, where $C$ is the number of classes in the target dataset. For example, with CIFAR-100, its size is $100 \times 1,000$. The total number of trainable parameters is relatively small and does not require gradients. This results in our method using less than 2GB of GPU memory.

In terms of computational complexity, we denote the batch size as $S$, encoder's output size as $D$, and class number as $C$. Therefore, the dimensions of $X$, $Y$, $R$ and $W$ are $S \times D$, $S \times C$, $D \times D$, and $C \times D$, respectively. Thus, the calculation is shown below:

The computational complexity for updating $R$ is dominated by the matrix multiplications, thus:

$$\max\{\mathcal{O}(SDC), \mathcal{O}(SC), \mathcal{O}(SDC)\} \approx \max\{\mathcal{O}(SDC), \mathcal{O}(D^2C)\}. \tag{14}$$

The computational complexity for updating $W$ is dominated by the matrix multiplications and the matrix inversion:

$$\max\{\mathcal{O}(SD), \mathcal{O}(SD^2), \mathcal{O}(S^2), \mathcal{O}(S^3), \mathcal{O}(DS^2), \mathcal{O}(D^2S), \mathcal{O}(D^2)\} \approx \max\{\mathcal{O}(S^3), \mathcal{O}(D^2S)\}. \tag{15}$$

In the OCIL setting, the batch size is relatively smaller. Therefore, the overall computational complexity is primarily controlled by $D$.

### 3.5 Overhead Analysis

In terms of space overhead, compared to the conventional backbone + classifier structure, F-OAL introduces an additional linear projection to control the output dimension $D$ of the encoder, and a matrix $R$, where only $R$ is trainable. According to Equation 9, the dimension of $R$ remains a fixed size of $D \times D$. Other methods require more extra space. For instance, LwF [20] employs knowledge distillation, necessitating the storage of additional models, while replay-based methods require extra storage to retain historical samples. In contrast, the overhead introduced by F-OAL, consisting of an additional matrix and a linear layer, is smaller.

| Metric | Method | Replay? | CIFAR-100 | CORe50 | FGCVAircraft | DTD | Tiny-ImageNet | Country211 |
|---|---|---|---|---|---|---|---|---|
| $A_{avg}(\%)\uparrow$ | iCaRL(CVPR 2017)[31] | ✓ | 91.6 | 95.6 | 36.4 | 74.1 | 91.3 | 12.2 |
| | ER(ICRA 2019)[12] | ✓ | 90.1 | 94.8 | 35.7 | 65.4 | 87.3 | 14.0 |
| | ASER(AAAI 2021)[33] | ✓ | 87.2 | 87.1 | 25.2 | 57.4 | 85.8 | 13.2 |
| | SCR(CVPR 2021)[25] | ✓ | 91.9 | 95.3 | 55.6 | 75.0 | 82.6 | 14.7 |
| | DVC(CVPR 2022)[9] | ✓ | 92.4 | 97.1 | 33.7 | 67.3 | 91.5 | 16.1 |
| | PCR(CVPR 2023)[21] | ✓ | 89.1 | 95.7 | 10.1 | 35.0 | 91.0 | 9.7 |
| | LwF(TPAMI 2018)[20] | ✗ | 69.3 | 47.0 | 14.2 | 40.2 | 82.5 | 1.4 |
| | EWC(PNAS 2017)[17] | ✗ | 49.9 | 47.9 | 12.0 | 27.6 | 60.5 | 6.1 |
| | EASE(CVPR 2024)[42] | ✗ | 91.1 | 85.0 | 38.2 | 76.0 | **92.0** | 15.9 |
| | LAE(ICCV 2023)[8] | ✗ | 79.1 | 73.3 | 13.5 | 63.5 | 86.7 | 14.5 |
| | SLCA(ICCV 2023)[40] | ✗ | 90.4 | 93.7 | 34.3 | 70.9 | 88.6 | 17.8 |
| | F-OAL | ✗ | **91.1** | **96.3** | **62.2** | **82.8** | 91.2 | **24.4** |
| $A_{last}(\%)\uparrow$ | iCaRL(CVPR 2017) | ✓ | 87.5 | 93.2 | 29.8 | 66.3 | 87.8 | 6.5 |
| | ER(ICRA 2019) | ✓ | 84.6 | 92.1 | 28.6 | 54.3 | 81.6 | 6.8 |
| | ASER(AAAI 2021) | ✓ | 82.0 | 82.1 | 14.8 | 49.4 | 80.0 | 6.7 |
| | SCR(CVPR 2021) | ✓ | 87.7 | 93.6 | 50.3 | 68.7 | 75.8 | 8.0 |
| | DVC(CVPR 2022) | ✓ | 87.8 | 96.0 | 27.0 | 57.2 | 87.2 | 9.2 |
| | PCR(CVPR 2023) | ✓ | 81.4 | 93.9 | 9.0 | 34.6 | 86.1 | 6.1 |
| | LwF(TPAMI 2018) | ✗ | 64.8 | 26.3 | 5.8 | 18.3 | 72.5 | 0.5 |
| | EWC(PNAS 2017) | ✗ | 25.2 | 21.1 | 3.0 | 13.3 | 44.6 | 1.9 |
| | EASE(CVPR 2024) | ✗ | 85.4 | 78.3 | 29.3 | 67.6 | **89.3** | 10.5 |
| | LAE(ICCV 2023) | ✗ | 75.6 | 67.1 | 6.3 | 53.6 | 82.4 | 9.3 |
| | SLCA(ICCV 2023) | ✗ | 85.6 | 88.2 | 32.1 | 63.3 | 85.4 | 12.9 |
| | F-OAL | ✗ | **86.5** | **92.5** | **54.0** | **75.9** | 87.3 | **17.5** |
| $F(\%)\downarrow$ | iCaRL(CVPR 2017) | ✓ | 3.2 | 2.3 | 7.1 | 7.8 | 2.7 | 6.7 |
| | ER(ICRA 2019) | ✓ | 20.7 | 4.3 | 34.0 | 29.8 | 13.3 | 21.0 |
| | ASER(AAAI 2021) | ✓ | 16.5 | 9.9 | 35.7 | 29.3 | 16.3 | 19.4 |
| | SCR(CVPR 2021) | ✓ | 6.2 | 3.8 | 14.5 | 11.6 | 7.7 | 6.4 |
| | DVC(CVPR 2022) | ✓ | 8.2 | 2.3 | 29.7 | 21.7 | 8.9 | 18.9 |
| | PCR(CVPR 2023) | ✓ | 9.2 | 4.2 | 2.7 | 1.4 | 8.0 | 1.7 |
| | LwF(TPAMI 2018) | ✗ | **1.3** | **0.4** | **3.1** | **4.5** | **1.0** | **0** |
| | EWC(PNAS 2017) | ✗ | 67.4 | 81.0 | 38.8 | 68.3 | 20.7 | 51.5 |
| | EASE(CVPR 2024) | ✗ | 6.1 | 10.7 | 19.2 | 12.5 | 2.8 | 16.8 |
| | LAE(ICCV 2023) | ✗ | 11.8 | 13.8 | 12.2 | 25.0 | 5.4 | 16.7 |
| | SLCA(ICCV 2023) | ✗ | 7.1 | 3.4 | 10.2 | 12.7 | 4.2 | 14.9 |
| | F-OAL | ✗ | 5.5 | 3.9 | 10.0 | 10.1 | 5.0 | 6.9 |

Table 2: This table shows the comparison results of our method with other baselines on six datasets. We select three metrics: average accuracy ($A_{avg}$), last task accuracy ($A_{last}$), and forgetting rate ($F$). Higher values for $A_{avg}$ and $A_{last}$ indicate better performance, while lower values for $F$ indicate better performance. In **Replay?** column, replay-based methods are marked with ✓. Conversely, exemplar-free methods are marked with ✗. Data in **Bold** are the best within exemplar-free methods, and data underlined are the best considering both categories.

In terms of time overhead, our method primarily consists of a forward pass and matrix multiplication, which is determined by the output dimension of the encoder. By changing the output dimension of the encoder, we can balance the accuracy and time. According to [16], the backward pass in back-propagation (forward pass + backward pass) accounts for 70% of the time. Therefore, our method's time overhead is also relatively small.

# 4 Experiment

To show the effectiveness of F-OAL, we conduct extensive experiments to compare our approach with baseline methods. We build our code and reproduce relevant results based on [24, 34]. All baselines are with pre-trained ViT as the backbone.

## 4.1 Datasets

We focus on coarse-grained data scenarios, such as CIFAR-100, Tiny-ImageNet, and Core50, as well as fine-grained data scenarios, including DTD, FGVCAircraft, and Country211. All of these are open-source benchmark datasets. However, there are other data scenarios, such as long-tail distributions, where unbalanced data distribution will make it harder to achieve good performance by only training the classifier. We will study this case in future work.

• **CIFAR-100** [18] is constructed into 10 tasks with disjoint classes, and each task has 10 classes. Each task has 5,000 images for training and 1,000 for testing.

- **CORe50** [23] is a benchmark designed for class incremental learning with 9 tasks and 50 classes: 10 classes in the first task and 5 classes in the subsequent 8 tasks. Each class has 2,398 training images and 900 testing images on average.

- **FGVCAircraft** [26] contains 102 different classes of *aircraft models*. 100 classes are selected and divided into 10 tasks. Each class has 33 training images and 33 testing images on average.

- **DTD** [3] is a *texture database*, organized into 47 classes. 40 classes are selected and divided into 10 tasks. Each class has 40 training images and 40 testing images.

- **Tiny-ImageNet** is a subset of ImageNet with 200 classes for training. Each class has 500 images. The test set contains 10,000 images. The dataset is evenly divided into 10 tasks.

- **Country211** contains 211 classes of country images, with 150 train and test images per class. The dataset is evenly divided into 10 tasks with 210 classes chosen.

## 4.2 Evaluation Metrics

We define $a_{i,j}$ as the accuracy evaluated on the test set of task $j$ after training the network from task 1 through to $i$, and the average accuracy is defined as

$$A_i = \frac{1}{i} \sum_{j=1}^{i} a_{i,j}. \tag{16}$$

When $i = m$ (i.e., the total number of tasks), $A_m$ represents the average accuracy by the end of training.

Forgetting rate at task $i$ is defined as Equation 17. $f_{i,j}$ represents how much the model has forgot about task $j$ after being trained on task $i$. Specifically, $\max_{l \in \{1,...,k-1\}} (a_{l,j})$ denotes the best test accuracy the model has ever achieved on task $j$ before learning task $k$, and $a_{k,j}$ is the test accuracy on task $j$ after learning task $k$.

$$F_i = \frac{1}{i-1} \sum_{j=1}^{i-1} f_{i,j}, \tag{17}$$

where

$$f_{k,j} = \max_{l \in \{1,...,k-1\}} (a_{l,j}) - a_{k,j}, \quad \forall j < k. \tag{18}$$

## 4.3 Implementation Details

ViT-B [5], pre-trained on ImageNet-1K [4], serves as the backbone network for all methods. The data stream is kept identical across all experiments to ensure a fair comparison. The learning rate and batch size are set to 0.001 and 10, respectively. The optimizer is SGD. We assign 5,000 memory buffer sizes for replay-based methods. In F-OAL, the expansion size is 1,000 (i.e., the output activation size to update AC is 1,000). The regularization term is set to be 1. All experiments are conducted on a single RTX 4070ti GPU 12GB, and an average of 3 runs is reported.

## 4.4 Result Comparison

We tabulate the average accuracy ($A_{avg}$), last task accuracy ($A_{last}$), and the forgetting rate ($F$) from the compared methods in Table 2.

For fine-grained datasets such as FGCVAircraft, DTD, and Country211, F-OAL achieves the best performance, with average accuracies of 66.2%, 82.8%, and 24.4%, respectively. The second-best results are 55.6%, 76.0%, and 17.8%, respectively. Similarly, F-OAL also outperforms in last task accuracy. This demonstrates the excellent transferability of our method in the OCIL setting, effectively leveraging the feature extraction capabilities of the pre-trained encoder.

On coarse-grained datasets such as CIFAR-100, CORe50, and Tiny-ImageNet, F-OAL still demonstrates excellent performance, achieving the highest accuracy among all exemplar-free methods, except on Tiny-ImageNet where it is 0.8% behind EASE in average accuracy. Compared to the best replay-based methods, it only lags by 1.3%, 0.8%, and 0.3%, respectively.

| Methods | CIFAR-100 | CORe50 | FGVCAircraft | DTD | Tiny-ImageNet | Country211 |
|---|---|---|---|---|---|---|
| LwF | 412 | 877 | 135 | 73 | 841 | 256 |
| EWC | 451 | 922 | 115 | 62 | 1,190 | 249 |
| iCaRL | 832 | 1,716 | 53 | 24 | 1671 | 513 |
| ER | 652 | 1,433 | 40 | 21 | 1,315 | 404 |
| ASER | 5,608 | 7,700 | 91 | 43 | 18,597 | 20,611 |
| SCR | 2,843 | 5,939 | 88 | 42 | 62,996 | 810 |
| DVC | 4,191 | 9,351 | 287 | 130 | 10,940 | 2,622 |
| PCR | 1,624 | 3,742 | 113 | 53 | 3,274 | 1,028 |
| EASE | 383 | 760 | 147 | 139 | 638 | 304 |
| LAE | **252** | **458** | 156 | 140 | **500** | 355 |
| SLCA | 726 | 1,416 | 289 | 278 | 1,185 | 551 |
| F-OAL | 261 | 570 | **16** | **8** | 507 | **157** |

Table 3: Training time including feature extraction is reported in seconds where replay-based methods are with 5,000 buffer size. Data in **Bold** show the fastest time.

Typically, a lower forgetting rate is better, but forgetting is based on accuracy. Therefore, when comparing forgetting rates, it is important to consider models with similar accuracy levels. When comparing F-OAL to the well-performing DVC, F-OAL exhibits a lower forgetting rate. On CIFAR-100, F-OAL's forgetting rate is 5.5%, while DVC's is 8.2%. This indicates that F-OAL not only maintains a high level of accuracy but also effectively manages forgetting.

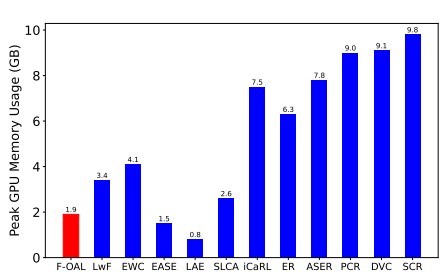

Figure 2: Peak GPU memory footprint in GB with 10 batch size on CIFAR-100. Replay-based methods are with 5,000 buffer size. F-OAL has low GPU footprint since it does not require gradients.

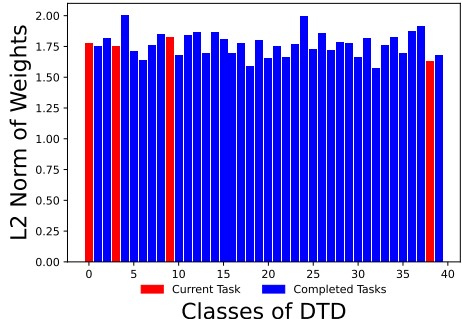

Figure 3: Visualization of the weights of a linear classifier. The result comes from F-OAL on DTD. Based on [15], when recency bias happens, L2 norm of current task is significantly larger. In F-OAL, the L2 norm of current task is in a average level.

### 4.5 Resource Consumption

**GPU**. Peak GPU memory footprint on CIFAR-100 is shown in Figure 2. As we state in Section 3.5, without using gradient and extra space for exemplars, F-OAL requires a low memory footprint while having good performance (i.e., higher accuracy than LAE and EASE on most datasets).

**Training Time**. Table 3 illustrates training time including feature extraction. F-OAL is fast with competitive accuracy (i.e., higher accuracy than LAE). Only the classifier and regularized feature autocorrelation matrix are updated, leading to fewer of trainable parameters and fast training speed.

### 4.6 Countering Recency Bias

As demonstrated in Figure 3, the linear classifier of AC obtained through F-OAL training does not exhibit recency bias. Notably, the weights corresponding to the classes in the most recent tasks do not significantly surpass those of the earlier classes. The frozen encoder and the recursive least square updating manner ensure equal treatment for all samples.

## 4.7 Ablation Study

We design the ablation study in Table 4 to verify the effectiveness of the Feature Fusion and Smooth Projection modules. Without these two components, the accuracy of F-OAL drops, especially on fine-grained datasets.

| FF | SP | CIFAR-100 | CORe50 | FGVCAircraft | DTD | Tiny-ImageNet | Country211 |
|----|----|-----------|--------|--------------|-----|---------------|------------|
| ✓  | ✓  | 91.1      | 96.3   | 62.2         | 82.8 | 91.2         | 24.4       |
| ✗  | ✓  | 90.6      | 95.3   | 60.9         | 80.5 | 91.4         | 21.3       |
| ✓  | ✗  | 90.7      | 95.4   | 58.7         | 79.3 | 91.2         | 22.8       |
| ✗  | ✗  | 90.6      | 95.4   | 56.0         | 71.2 | 91.4         | 21.1       |

Table 4: Average accuracy comparison with Feature Fusion (FF) and Smooth Projection (SP) modules across various datasets. ✓ means **with** and ✗ means **without**.

As Table 5 (See appendix B) shows, we prove analytic classifier is key to high accuracy. Herein, we define the Fully Connected Classifier (FCC) as having the identical structure to the AC, but it is updated through back-propagation rather than utilizing the $R$ and recursive least square. Without AC, the accuracy drops from 91.1% to 32.4% on CIFAR-100.

**Regularization Term.** Table 6 (See appendix B) shows the effects of varying $\gamma$. For large volume dataset, F-OAL gives a robust performance in wide range of $\gamma$ value (e.g., $10^2$ - $10^{-3}$), while small datasets need larger value (e.g., $10^2$ - 1). The comprehensive results show that $\gamma$=1 is the suitable choice.

**Projection Size**. Figure 4 (See appendix B) demonstrates the influence of different random projection sizes. The results suggest that the setting of a 1,000-dimensional projection is appropriate.

## 4.8 Potential Positive and Negative Societal Impacts

The key advantage of our approach is its ability to achieve exemplar-free OCIL with fast training and low memory footprint, offering an environmentally friendly and efficient solution for this research track. However, the main limitation of our method lies in its reliance on a powerful pre-trained encoder. As a result, it is crucial for us to leverage open-source pre-trained backbone networks from the deep learning community, rather than training our own, which will otherwise lead to higher GPU usage and increased resource consumption.

# 5 Conclusion

In this paper, we propose Forward-only Online Analytic Learning (F-OAL), an exemplar-free method for addressing several limitations of the Online Class Incremental Learning scenario. We use Analytic Learning to acquire the optimal solution of the classifier directly instead of training for dozens of epochs by back-propagation. Leveraging the frozen pre-trained encoder with Feature Fusion and Smooth Projection and the Analytic Classifier updated by recursive least square, our approach achieves the identical solution to joint-learning on the whole dataset without preserving any historical exemplars, achieving high accuracy and reducing the resource consumption. Our experiments show the competitive performance of F-OAL.

# 6 Acknowledgment

This research was supported by the National Natural Science Foundation of China (62306117), the Guangzhou Basic and Applied Basic Research Foundation (2024A04J3681, 2023A04J1687), the South China University of Technology-TCL Technology Innovation Fund, the Fundamental Research Funds for the Central Universities (2023ZYGXZR023, 2024ZYGXZR074), the Guangdong Basic and Applied Basic Research Foundation (2024A1515010220), and the CAAI- MindSpore Open Fund developed on Openl Community.

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

# A  Appendix

**Proof of Theorem 3.1 and 3.2**

For **Theorem 3.1**, we start with proving the case in batch 1 of task $k$.

According to Equation 8, of task $k-1$, we can expand the stacked activation and label matrixes:

$$\hat{W}^{(k-1,n)} = \left( \sum_{m=1}^{k-1} \sum_{i=1}^{n} X_{m,i}^{(a)T} X_{m,i}^{(a)} + \gamma I \right)^{-1} \begin{bmatrix} X_{1,1}^{(a)T} Y_{1,1}^{train} \\ \vdots \\ X_{k-1,n}^{(a)T} Y_{k-1,n}^{train} \end{bmatrix}. \tag{19}$$

Hence, at batch 1 of task $k$, we have

$$\hat{W}^{(k,1)} = \left( \sum_{m=1}^{k-1} \sum_{i=1}^{n} X_{m,i}^{(a)T} X_{m,i}^{(a)} + \gamma I + X_{k,1}^{(a)T} X_{k,1}^{(a)} \right)^{-1} \begin{bmatrix} X_{1,1}^{(a)T} Y_{1,1}^{train} \\ \vdots \\ X_{k-1,n}^{(a)T} Y_{k-1,n}^{train} \\ X_{k,1}^{(a)T} Y_{k,1}^{train} \end{bmatrix}. \tag{20}$$

We have defined regularized feature autocorrelation matrix $R_{k-1,n}$ via

$$R_{k-1,n} = \left( \sum_{m=1}^{k-1} \sum_{i=1}^{n} X_{m,i}^{(a)T} X_{m,i}^{(a)} + \gamma I \right)^{-1} \tag{21}$$

To facilitate subsequent calculations, here we also define a cross-correlation matrix $Q_{k-1,n}$

$$Q_{k-1,n} = \begin{bmatrix} X_{1,1}^{(a)T} Y_{1,1}^{train} & \cdots & X_{k-1,n}^{(a)T} Y_{k-1,n}^{train} \end{bmatrix}. \tag{22}$$

Thus, we can rewrite Equation 20 as

$$\hat{W}^{(k-1,n)} = R_{k-1,n} Q_{k-1,n}. \tag{23}$$

Therefore, at batch 1 of task $k$ we have

$$\hat{W}^{(k,1)} = R_{k,1} Q_{k,1}. \tag{24}$$

From Equation 21 we can recursively calculate $R_{k,1}$ from $R_{k-1,n}$

$$R_{k,1} = \left( R_{k-1,n}^{-1} + X_{k,1}^{(a)T} X_{k,1} \right)^{-1}. \tag{25}$$

According to the Woodbury matrix identity, we have

$$(A + UCV)^{-1} = A^{-1} - A^{-1} U \left( C^{-1} + VA^{-1}U \right)^{-1} VA^{-1}. \tag{26}$$

Let $A = R_{k-1,n}^{-1}, U = X_{k,1}^{(a)T}, C = I, V = X_{k,1}^{(a)}$ in Equation 26, we have

$$R_{k,1} = R_{k-1,n} - R_{k-1,n} X_{k,1}^{(a)T} \left( I + X_{k,1}^{(a)} R_{k-1,n} X_{k,1}^{(a)T} \right)^{-1} X_{k,1}^{(a)} R_{k-1,n}. \tag{27}$$

Hence, $R_k$ can be recursively updated using its last task counterpart $R_{k-1,n}$ and current data (i.e., $X_{k,1}$). This proves the recursive calculation in Equation 11.

Next, we derive the recursive formulation of $\hat{W}^{(k,1)}$. To this end, we also recuse the cross-correlation matrix $Q_{k,1}$ at the batch 1 of task $k$:

$$Q_{k,1} = \begin{bmatrix} X_{1,1}^{(a)T} Y_{1,1}^{train} & \cdots & X_{k-1,n}^{(a)T} Y_{k-1,n}^{train} & X_{k,1}^{(a)T} Y_{k,1}^{train} \end{bmatrix} = Q'_{k-1,n} + X_{k,1}^{(a)T} Y_{k,1}^{train}. \tag{28}$$

where

$$Q'_{k-1,n} = \begin{bmatrix} Q_{k-1,n} & 0_{d_{(a)} \times d_{yk}} \end{bmatrix} \tag{29}$$

where $d_{(a)}$ is the dimension of activation and $d_{yk}$ is dimension of total classes of task $k$. Note that the concatenation in Equation 29 is due to fact that $Y_{k,1}$ of task $k$ contains more data classes (hence more columns) than $Y_{k-1,n}$

Let $K_{k,1} = \left( I + X_{k,1}^{(a)} R_{k-1,n} X_{k,1}^{(a)T} \right)^{-1}$. Since $I = K_{k,1} K_{k,1}^{-1} = K_{k,1} \left( I + X_{k,1}^{(a)} R_{k-1,n} X_{k,1}^{(a)T} \right)$, we have $K_{k,1} = I - K_{k,1} X_{k,1}^{(a)} R_{k-1,n} X_{k,1}^{(a)T}$. Therefore,

$$\begin{aligned}
\boldsymbol{R}_{k-1,n}\boldsymbol{X}_{k,1}^{(a)T}(\boldsymbol{I}+\boldsymbol{X}_{k,1}^{(a)}\boldsymbol{R}_{k-1,n}\boldsymbol{X}_{k,1}^{(a)T})^{-1} &= \boldsymbol{R}_{k,1}\boldsymbol{X}_{k,1}^{(a)T}\boldsymbol{K}_{k,1}\\
&= \boldsymbol{R}_{k,1}\boldsymbol{X}_{k}^{(a)T}(\boldsymbol{I}-\boldsymbol{K}_{k,1}\boldsymbol{X}_{k,1}^{(a)}\boldsymbol{R}_{k-1,n}\boldsymbol{X}_{k,1}^{(a)T})\\
&= (\boldsymbol{R}_{k-1,n}-\boldsymbol{R}_{k-1,n}\boldsymbol{X}_{k,1}^{(a)T}\boldsymbol{K}_{k,1}\boldsymbol{X}_{k,1}^{(a)}\boldsymbol{R}_{k-1,n})\boldsymbol{X}_{k}^{(a)T}\\
&= \boldsymbol{R}_{k,1}\boldsymbol{X}_{k,1}^{(a)T}.
\end{aligned} \tag{30}$$

As $\hat{\boldsymbol{W}}^{(k-1,n)'}=[\,\hat{\boldsymbol{W}}^{(k-1,n)}\quad \boldsymbol{0}]$ has expanded its dimension similar to what $\boldsymbol{Q}'_{k-1,n}$ does, we have

$$\hat{\boldsymbol{W}}^{(k-1,n)'}=\boldsymbol{R}_{k-1,n}\boldsymbol{Q}'_{k-1,n}. \tag{31}$$

Hence, $\hat{\boldsymbol{W}}^{(k,1)}$ can be rewritten as

$$\hat{\boldsymbol{W}}^{(k,1)}=\boldsymbol{R}_{k,1}\boldsymbol{Q}_{k,1}=\boldsymbol{R}_{k,1}(\boldsymbol{Q}'_{k-1,n}+\boldsymbol{X}_{k,1}^{(a)T}\boldsymbol{Y}_{k,1}^{\text{train}})=\boldsymbol{R}_{k,1}\boldsymbol{Q}'_{k-1,n}+\boldsymbol{R}_{k,1}\boldsymbol{X}_{k,1}^{(a)T}\boldsymbol{Y}_{k,1}^{\text{train}}. \tag{32}$$

By substituting Equation 27 into $\boldsymbol{R}_{k,1}\boldsymbol{Q}'_{k-1,n}$, we have

$$\begin{aligned}
\boldsymbol{R}_{k,1}\boldsymbol{Q}'_{k-1,n} &= \boldsymbol{R}_{k-1,n}\boldsymbol{Q}'_{k-1,n}-\boldsymbol{R}_{k-1,n}\boldsymbol{X}_{k,1}^{(a)T}(\boldsymbol{I}+\boldsymbol{X}_{k,1}^{(a)}\boldsymbol{R}_{k-1,n}\boldsymbol{X}_{k,1}^{(a)T})^{-1}\boldsymbol{X}_{k,1}^{(a)}\boldsymbol{R}_{k-1,n}\boldsymbol{Q}'_{k-1,n}\\
&= \boldsymbol{W}^{(k-1,n)'}-\boldsymbol{R}_{k-1,n}\boldsymbol{X}_{k,1}^{(a)T}(\boldsymbol{I}+\boldsymbol{X}_{k,1}\boldsymbol{R}_{k-1,n}\boldsymbol{X}_{k,1}^{(a)T})^{-1}\boldsymbol{X}_{k,1}^{(a)}\boldsymbol{W}^{(k-1,n)'}.
\end{aligned} \tag{33}$$

According to Equation 30, Equation 33 can be rewritten as

$$\boldsymbol{R}_{k,1}\boldsymbol{Q}'_{k-1,n}=\hat{\boldsymbol{W}}^{(k-1,n)'}-\boldsymbol{R}_{k,1}\boldsymbol{X}_{k,1}^{(a)T}\boldsymbol{X}_{k,1}^{(a)}\hat{\boldsymbol{W}}^{(k-1,n)'}. \tag{34}$$

By inserting Equation 33 into Equation 32, we have

$$\begin{aligned}
\hat{\boldsymbol{W}}^{(k,1)} &= \hat{\boldsymbol{W}}^{(k-1,n)'}-\boldsymbol{R}_{k,1}\boldsymbol{X}_{k,1}^{(a)T}\boldsymbol{X}_{k,1}^{(a)}\hat{\boldsymbol{W}}^{(k-1,n)'}+\boldsymbol{R}_{k,1}\boldsymbol{X}_{k,1}^{(a)T}\boldsymbol{Y}_{k,1}^{\text{train}}\\
&= \hat{\boldsymbol{W}}^{(k-1,n)'}+\boldsymbol{R}_{k,1}\boldsymbol{X}_{k,1}^{(a)T}(\boldsymbol{Y}_{k,1}^{\text{train}}-\boldsymbol{X}_{k,1}^{(a)}\hat{\boldsymbol{W}}^{(k-1,n)'}),
\end{aligned} \tag{35}$$

which proves the case in the batch 1 of task $k$.

For **Theorem 3.2**, we consider the case at rest batches of task $k$. In the rest of batches, the number of classes maintain unchanged compared with batch 1, which means no column expansion is required. According to Equation 35, we substitute $\hat{\boldsymbol{W}}^{(k-1,n)'}$ by $\hat{\boldsymbol{W}}^{(k,i-1)}$. Similar substitution is applied to $\boldsymbol{R}_{k-1,n}$ with $\boldsymbol{R}_{k,i-1}$ according to Equation 27 since the shape of regularized feature autocorrelation matrix is remained through whole learning agenda. Then we have

$$\hat{\boldsymbol{W}}^{(k,i)}=\hat{\boldsymbol{W}}^{(k,i-1)}+\boldsymbol{R}_{k,i}\boldsymbol{X}_{k,i}^{(a)T}\left(\boldsymbol{Y}_{k,i}^{\text{train}}-\boldsymbol{X}_{k,i}^{(a)}\hat{\boldsymbol{W}}^{(k,i-1)}\right), \tag{36}$$

$$\boldsymbol{R}_{k,i}=\boldsymbol{R}_{k,i-1}-\boldsymbol{R}_{k,i-1}\boldsymbol{X}_{k,i}^{(a)T}\left(\boldsymbol{I}+\boldsymbol{X}_{k,i}^{(a)}\boldsymbol{R}_{k,i-1}\boldsymbol{X}_{k,i}^{(a)T}\right)^{-1}\boldsymbol{X}_{k,i}^{(a)}\boldsymbol{R}_{k,i-1}, \tag{37}$$

which complete the proof. $\qquad\square$

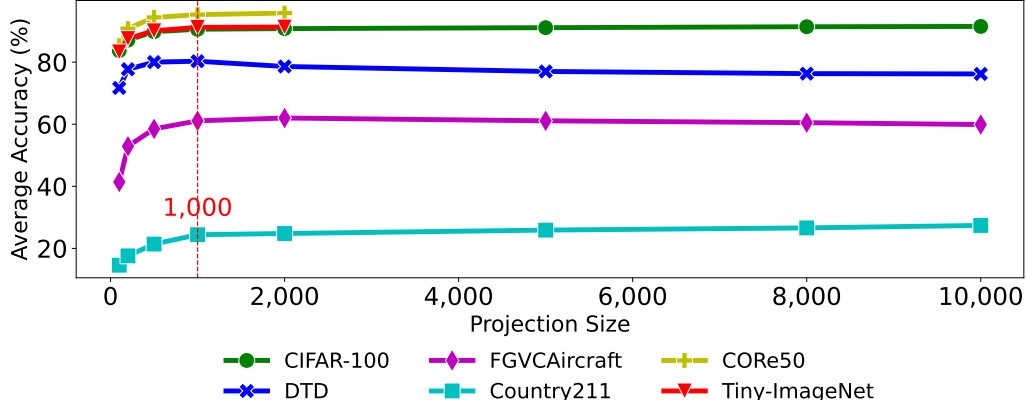

Figure 4: Average accuracy measured in different projection sizes. Due to the time complexity in computing Equation 11, higher projection sizes escalate training time while increase little performance. On small datasets, higher projection sizes may even lead to reduction.

## B    Appendix

Figure 4 demonstrates the influence of different random projection sizes. These experiments reveal that higher-dimensional projections may not always improve accuracy, suggesting that projection size must fit the dataset's volume. If the volume of the dataset is small, a high-dimensional projection results in over-fitting to the training set. The maximum accuracy on DTD is at 1,000 projection size, while the accuracy slightly increases after 1,000. When the projection size is 2,000, the results of FGVCAircrft and CORe50 are a little higher than the results on 1,000, but time consumption is twice higher due to the cubic time complexity in the matrix inverse. Thus, we limit the projection size measurement to 2,000 for CORe50 and Tiny-ImageNet due to the significant increase in time consumption.

Table 5 justify the contribution of AC. The reduction from 91.1% to 32.4% without AL suggesting that one epoch of simple back-propagation is unable to handle the OCIL.

Table 6 varifies the choice of 1 as regularization term is suitable for OCIL. Fine-grained datasets suffer a lot from the wrong choice of $\gamma$. The average accuracy on FGVCAircraft drops from 61.1% to 22.4% when $\gamma$ is tuned from 1 to $10^{-3}$.

| AC | FCC | Result |
|:---:|:---:|:---:|
| ✓ | ✗ | **91.1** |
| ✗ | ✓ | 32.4 |

Table 5: Average accuracy of CIFAR-100 with updating manner in classifier.

| Dataset | $10^2$ | $10^1$ | 1 | $10^{-1}$ | $10^{-2}$ | $10^{-3}$ |
|:---|:---:|:---:|:---:|:---:|:---:|:---:|
| CIFAR-100 | 91.1 | 91.1 | **91.1** | 90.8 | 90.3 | 87.2 |
| CORe50 | 95.3 | 95.3 | **95.3** | 95.2 | 93.6 | 89.9 |
| FGVCAircraft | 55.1 | 59.5 | **61.1** | 47.1 | 34.9 | 22.4 |
| DTD | 70.3 | 75.2 | **81.6** | 81.5 | 79.1 | 70.8 |
| Tiny-ImageNet | 91.2 | 91.2 | **91.2** | 90.9 | 90.3 | 89.8 |
| Country211 | 23.2 | 23.3 | **24.4** | 22.6 | 21.3 | 17.4 |

Table 6: Average accuracy measured with different regularization terms. $\gamma$=1 shows robust results across all datasets.

