# OpenReview forum: "F-OAL: Forward-only Online Analytic Learning with Fast Training and Low Memory Footprint in Class Incremental Learning"
_NeurIPS.cc/2024/Conference — NeurIPS 2024 poster_

### Official Review · Reviewer_XB6P · 2024-07-04

**Soundness:** 2
**Presentation:** 2
**Contribution:** 2
**Rating:** 4
**Confidence:** 4

**Summary:**

The author proposed a method called F-OAL for online class incremental learning, which does not rely on back-propagation and is forward-only, significantly reducing memory usage and computational time. In summarize, the contributions are as follows:
1) The paper presents theF-OAL, which is an exemplar-free method.
2) F-OAL can be updated in an mini-batch manner;
3) The methods are evaluated in several benchmarks;

**Strengths:**

Strengths:
The author proposed a method called F-OAL for online class incremental learning, which does not rely on back-propagation and is forward-only, significantly reducing memory usage and computational time.

**Weaknesses:**

1) Some of the descriptions are unclear, such as Formula 4. The author may want to give a more vivid explanation;
2) The innovation is limited, See Limitations No 5;
3) The author may want to discuss more methods in evulation part;

**Questions:**

See Limitations.
The author may want to addresses the questions prpopsed in Limitations.

**Limitations:**

1. The formula No.4 seems to be wrong. Please check the formula carefully and give the proof process to ensure that the equation is valid.
2. Why does the formula No.4 show the optimal solution? For each mini-batch of data, the parameter W is the solution that makes the formula No.5 always equal to 0, in theory. But considering all batches of data, parameter W is too idealized and may even overfit the data of the current batch.
3. It is difficult for the parameter W calculated using a mini-batch of data to have an effect on other batches of data, especially when the distribution of data for different tasks is significantly different. Even if all the parameters W of the mini-batch are combined, I don’t think it can exceed the back-propagation based method in terms of effect, because this combination is linear.
4. Is the optimization process recursive on all batches of data? Why is the recursive method more efficient? Less computation? GPU parallel computing? Please explain the reason.
5. The innovation of this paper is insufficient. It seems that the main contribution is to calculate the parameter W by using the forward process and the least squares method. In fact, this method faces many disadvantages, such as overfitting.
6. Many models that appeared in the comparative experiments did not have annotated references, such as DVC in Table 1.

---

> ### Author Rebuttal · Authors · 2024-08-04
>
> # Response to Reviewer XB6P
>
> Thank you for the thorough review. We provide more detailed response below. Hope this can help you with your concerns.
>
> ## W1&L1: The formula No.4 seems to be wrong.
>
> Thank you for pointing out the typo. We revise $ϕ(X)Y$ to $ϕ(X)^⊤Y$ in equation 4.
>
> ## L2: Why does the formula No.4 show the optimal solution? Is W too idealized and may even overfit the data of the current batch?
>
> We appologize that our way of delivering the analytical solution can be misleading. In fact, Eq. 4 is the optimal LS solution to loss function in Eq. 3 in the non-CIL sense (like a preliminary **in the case with all the data**). That is, $W$ here **does NOT indicates the weight for any batch**, but for the one computed using **all data**.
>
> On the other hand, Eq. 5 indicates **the same loss function** as that of Eq. 3, but in the sense of breaking the dataset into $k$ segments (hence the "$1:k$" symbols). This leads to the least-squares solution in Eq. 7 (the same as Eq. 4 but in the sense of breaking the dataset into $k$ segments). Here the **optimal solution** $W$ does not guarantee Eq. 5 being 0. The "optimal" here indicates that the obtained $W$ allows the objective function in Eq. 5 to be the smallest (not necessarily 0 though).
>
> The use of Eq. 5 (with many "$1:k$") prepares us for the subsequent recursive derivations (i.e., from "$1:k-1$" to "$1:k$"). Based on this, we are able to derive the F-OAL mainly indicated by Eq. 10 and Eq. 12 in **recursive form**. Note that the $W$ recursively computed using  Eq. 10 or Eq. 12, obtains **the identical result to that of Eq. 7**. That is, the update formula in Eq. 10 or Eq. 12 considers both the current-batch data (e.g., $k$) and the past-batch data (e.g., $1:k-1$). In a sense, the F-OAL does no suffer from catastrophic forgetting (CF) at all!
>
> ## L3: Can recursive least square exceed the back-propagation?
>
> Our F-OAL beats BP because it does not suffer from CF (see analysis in L2). According to [1], using BP to update the model is the reason why CF happens, which leads to incomplete feature learning and recency bias [2]. Existing baselines are still based on BP and manage to alleviate CF. We redefine the OCIL into a recursive learning problem to avoid BP. According to [3], the recursive update paradigm could obtain results that are identical to their joint-learning counterparts.
>
> To give further proof, we provide a quick experiment on MNIST to show that BP suffers from CF, while recursive methods do not. A two-layer MLP is trained on first 5 classes of MNIST (i.e., number 0 to 4) via BP and we test it on these 5 classes. Then the MLP is trained on rest of the 5 classes (i.e., number 5 to 9) via BP incrementally for each class per phase and we test it on the old 5 classes (i.e., number 0 to 4). We apply the identical experiment on our F-OAL using the same MLP. The results are reported below. Please check the source code of the quick experiment in **General Response**.
>
> |         | Acc on 0:4 after trained on 0:4 | Acc on 0:4 after trained on 0:9 |
> |---------|--------------------------|--------------------------|
> | BP      | 98.4                     | 43.2                     |
> | F-OAL     | 98.2                     | 96.4                     |
>
> [1] Online continual learning through mutual information maximization, ICML 2022
>
> [2] Learning a unified classifier incrementally via rebalancing, CVPR 2019
>
> [3] Blockwise recursive Moore–Penrose inverse for network learning. IEEE TMSC-S, 2021
>
> ## L4: Is the optimization process recursive on all batches of data? Why is the recursive method more efficient?
>
> Yes, the optimization process is  on all batches of data. For **space efficiency**, our method is forward-only and exemplar-free, eliminating the need for gradients and additional memory buffers, thereby significantly reducing the GPU footprint.
>
> For **computational efficiency**, F-OAL does not have backward pass, resulting in a faster training. Accroding to [1], in BP, the backward pass spends 70% of the entire time.
>
> [1] Decoupled Parallel Backpropagation with Convergence Guarantee. ICML 2018
>
> ## L5: The innovation of this paper is insufficient. This method faces many disadvantages, such as overfitting.
>
> Our main contributions are 1) pinpoint BP as the main cause of CF, 2) introducing F-OAL, the recursive method instead of BP that well handles OCIL problem, and 3) fusion module and smoothed projection that enhance the performance of F-OAL.
>
> Regarding the overfitting disadvantage, we must respectably disagree. The F-OAL is mainly linear regression, which usually invites "under-fitting" instead of "over-fitting" in nature. In addition, we avoid possible over-fitting (due to small data), we have introduced an $L_{2}$ regularization in Eq. 5. However, the F-OAL needs a fixed backbone. This is an existing disadvantage that will be discussed.
>
>
> ## L6：Many models that appeared in the comparative experiments did not have annotated references, such as DVC in Table 1.
>
> Thank you for pointing out the issue. We will add citations for each method in Table 1 and also review the rest of the paper to ensure there are no other similar problems. The table 1 will be revised in the following form with the right citation:
>
> | Metric | Method                 | CIFAR-100 | CORe50 | FGVCAircraft | DTD  | Tiny-ImageNet | Country211 |
> |--------|------------------------|-----------|--------|--------------|------|---------------|------------|
> | $A_{avg}$(↑)  | DVC(CVPR 2024) [1]     |92.4| 97.1 |33.7 |67.3 |91.5 |16.1|
> ...
>
> [1] Not just selection, but exploration: Online class-incremental continual learning via dual view consistency. In CVPR 2022.
>
> Based on these additional results and clarifications, we hope you could consider increasing your score in support of this work. If not, could you kindly let us know what additionally needs to be done in your assessment to make this work ready for publication?

---

### Official Review · Reviewer_UyXG · 2024-07-10

**Soundness:** 3
**Presentation:** 2
**Contribution:** 3
**Rating:** 5
**Confidence:** 3

**Summary:**

This paper presents an analytic class incremental learning method that does not need backpropogation. The main idea is to use a pre-trained mode to extract features followed by random projection to higher dimensional space, and then use recursive least squares to update the linear regression weights. By doing so, the closed-form solution solves for all seen data thus guarantees no forgetting. The experiments in several class incremental image classification tasks show superior results than many continual learning baselines.

**Strengths:**

- The empirical results of the method is very strong compared to other continual learning baselines.

**Weaknesses:**

- Starting from section 3.2, it's better to give more precise definition for all notations, such as their dimensionality.
- There are some confusing notations that can be improved. E.g., at line 121 you use $k$ to denote task, but at line 123 you use $k$ to denote batch, then in equation 9 you use change to $n$ to denote batch.
- There are some writing issues like typos in the main, e.g. line 17, line 117, equation 4 (should be $\phi(X)^\top Y$?)

**Questions:**

- Does the time recorded in Table 2 for F-OAL include the ViT feature extraction time?

**Limitations:**

The limitations of the method are not discussed.

---

> ### Author Rebuttal · Authors · 2024-08-04
>
> # Response to Reviewer UyXG
>
> Thank you for your valuable time in reviewing. We provide detailed information for your concerns below
>
> ## W1: Give more precise definition for all notations, such as their dimensionality.
>
> Thank you for your suggestion. We have included a following notation table.
>
> | Name          | Description                                                      | Dimension                                      |
> |---------------|------------------------------------------------------------------|------------------------------------------------|
> | $ϕ(X)$         | Activation of all images                                         | $V×D$ ($V$ is the number of all images, $D$ is the encoder output dimension)  |
> | $Y$             | One-hot label of all images                                      | $V×M$ ($M$ is the number of all classes)                               |
> | $\hat{W}$       | Joint-learning result of classifier's weight matrix              | $M×D$                                           |
> | $X_{k,n}^{(a)}$ | Activation matrix of the n-th batch of the k-th task               | $S×D$ ($S$ is the batch size)                      |
> | $Y_{k,n}^{train}$| One-hot label matrix of the n-th batch of the k-th task            | $S×C_s$ ($C_s$ is the number of classes seen so far)|
> |$ X_{k,1:n}^{(a)}$| Activation matrix from the start to the n-th batch of the k-th task | $V_s×D$ ($V_s$ is the number of images seen so far)|
> | $Y_{k,1:n}^{train}$   | One-hot label matrix from the start to the nth batch of the k-th task | $V_s×C_s$ |
> | $\hat{W}^{(k,n)}$  | Classifier of the nth batch of the k-th task           | $C_s×D$                                        |
> | $R_{k,n}$       | Regularized feature autocorrelation matrix to the nth batch of the k-th task | $D×D$                                           |
>
>
> ## W2: There are some confusing notations that can be improved.
>
> Thank you for your comments. We will revise the confusing notions as instructed. For instance, we will use $k$ to represent the task index and $n$ to represent the batch index respectively to avoid confusion.
>
> ## W3: There are some writing issues like typos.
>
> Thank you for pointing out these writing issues! We have corrected "class" to "classes" and "plan" to "plans" in line 17, changed "a" to "an" in line 117, and revised $ϕ(X)Y$ to $ϕ(X)^{T}Y$ in equation 4. Additionally, we will carefully review the paper to correct any other typos.
>
> ## Q1: Does the time recorded in Table 2 for F-OAL include the ViT feature extraction time?
>
> Yes, we have documented the entire training process, from the image entering the model to the completion of the model updates.
>
> ## L1: The limitations of the method are not discussed.
>
> Thank you for the suggestion. The major limitation is that, our method relies on well pre-trained backbones such as ViT and ResNet pre-trained on ImageNet. However, there are many open-source pre-trained backbones available within the deep learning community, which are relatively easy to obtain. Additionally, leveraging pre-trained models for fine-tuning downstream tasks has become a mainstream approach. Hence, although the need for backbones is an existing limitation of F-OAL, it is feasible and in line with mainstream.

---

> > ### Comment · Reviewer_UyXG · 2024-08-11
> >
> > I appreciate the authors' rebuttal. I still have minor concerns on the paper's presentation, as well as the limitation that it relies on a strong pretrained encoder. I will consider adjusting my score during the closed discussion. Thanks!

---

> > > ### Author Response · Authors · 2024-08-12
> > > **Thank you for the response**
> > >
> > > Thank you for taking the time to read our response! This means a lot to us! Please let us know if there is anything more needed from us!

---

### Official Review · Reviewer_nHHU · 2024-07-17

**Soundness:** 3
**Presentation:** 3
**Contribution:** 2
**Rating:** 8
**Confidence:** 4

**Summary:**

The authors address the problem of online class incremental learning (OCIL), where new tasks arrive periodically in a data stream and the trainer seeks to learn these new tasks without catastrophic forgetting of past performance. The paper presents two modes of OCIL; replay-based methods and exemplar-free methods. Replay-based methods offer strong performance but require storing some amount of replay data from the stream to include in incremental training. Exemplar-free methods lift this limitation, but have not thus far achieved comparable performance to replay-based methods. The authors method (forward-only online analytic learning, or F-OAL) uses projections of blocks from a frozen, pre-trained ViT encoder as input to a trainable linear classifier, and then perform recursive least-squares updates on the linear classifier to avoid the catastrophic forgetting that backpropagation would cause in the same setting. The authors extensively compare their method by accuracy, training time, and memory usage to other recently-SOTA methods for OCIL, both replay-based and exemplar-free. The comparisons show that F-OAL significantly improves over existing exemplar-free methods and achieves comparable performance with replay-based methods, while being significantly more efficient in terms of both training time and memory usage. Lastly, the authors ablate their solution and find that keeping the ViT encoder frozen and using their analytically-learned classifier (instead of learning it through backpropogation) are both key ingredients in F-OAL's accuracy.

**Strengths:**

The paper is generally quite strong. The F-OAL method is natural and intuitive; overall, it seems like a significant improvement to the Pareto frontier of accurate and efficient OCIL. The comparisons to past SOTA in both replay-based and exemplar-free methods are extensive and compelling. The authors' exposition is clear and logical, and the experiments are largely informative and useful for the reader. The F-OAL method seems pragmatic, and a natural baseline to which all future work in this area should be compared.

**Weaknesses:**

This paper (and past papers in this vein) suggest that exemplar-free methods are "good for data privacy", but there is very little justification for this claim. I understand the basic premise as this: if replay-based methods force you to store some subset of the data stream, which is worse for users' data privacy than exemplar-free methods that don't require such storage. While it's worth noting as a design consideration/feature, I do not agree with this framing as relevant for "data privacy". There is no legitimate security model of privacy in the literature that would recognize this as "higher privacy". Once the model provider has seen/processed the data by running it through the model, any legitimate security model would view this as data that's been made public. Methods that enhance security (e.g. SMPC, HE, TEEs) and methods that provably reduce statistical privacy leakage (i.e. differential privacy) are orthogonal; these methods can be used interchangeably with both replay-based and exemplar-free methods! In a realistic setting, I can see how exemplar-free methods might help assuage compliance concerns or company-specific rules, but I know of no data privacy regulation or cryptographic threat model where this would be a relevant factor. I would suggest that the authors attempt to correct this misconception in the literature by simply stating the feature in terms of its utility: data need not be stored for replay in production. The reader can implicitly understand that this can have several benefits depending on the circumstances of the deployment.

Otherwise, the only criticism I'd have for the paper is its ablation study. While it's a useful sanity check to see their result of ablating AC and FCC, the results with Frozen are obvious and unnecessary. The ViT-B model was developed for ImageNet-sized datasets, of course it will overfit CIFAR-100! I think there are more useful ablations that could be performed (more on that below).

**Questions:**

It seems clear to me that the authors method of fusing the ViT blocks $B_i(.)$ with random linear projection is valuable, but also not entirely necessary. The goal of this approach appears two-fold; (1) capture information from different levels of abstraction in the representation that their analytic classifier uses, and (2) be able to control the dimensionality of that representation, which will surely need to be tune-able at training time for different datasets (e.g. to avoid overfitting). The ablation in Appendix B reassures the reader that (2) is necessary, but none of the ablations suggest that (1) is necessary. A simple ablation that could've helped would be to compare their block-averaging + smoothed projection approach with a simpler method that applies the smoothed projection to the last ViT block. I'd be curious to know why the authors chose this particular approach. The paper itself states that this feature fusion was implemented to "further enhance the representativeness of the features", but is there any work or experiment they can point to that suggests this?

In any case, this seems to be the only weak point of the paper, and a clarification or improvement would be nice.

**Limitations:**

The only concern I would have is that the paper (and its cited works) abuse the notion of "data privacy", which I have previously argued against in this review. Works that claim to "improve data privacy" without treatment of the staggering amount of literature that has gone into defining and proving what is and is not "private" in an information-theoretic, statistical, or engineering-focused sense are likely to muddy the waters for those fields. The paper's setting and solution are already worth publishing; in my opinion, the unsubstantiated data privacy claim is hurting more than it's helping.

---

> ### Author Rebuttal · Authors · 2024-08-04
>
> # Response to Reviewer nHHU
>
> Thank you for your positive reviews and helpful suggestions. We provide detailed responses to your concerns below.
>
> ## W1: Abuse the notion of "data privacy".
>
> Thank you for raising this important concern. We agree that the use of “data privacy” is less appropriate in this paper. We will remove the use of "data privacy" and stick to “exemplar-free” (i.e., data need not be stored for replay in production).
>
> ## Q1: Ablation study results with frozen are obvious and unnecessary.
>
> Thank you for the suggestion. We will remove this unnecessary experiment and add more important ablation experiments based on your question.
>
> ## Q2: Simple ablation that could've helped would be to compare their block-averaging + smoothed projection approach.
>
> Thank you for pointing out the missing items in our ablation study. We have included a new set of experiments as follows according to your suggestions. The results are shown below.
>
> | Block-averaging| Smoothed Projection | CIFAR-100 | CORe50 | FGVCAircraft | DTD  | Tiny-ImageNet | Country211 |
> |--------|-----|------|--------|--------------|------|---------------|------------|
> | √   | √  | 91.1      | 96.3   | 62.2         | 82.8 | 91.2          | 24.4       |
> | ×   |   √| 90.6      | 95.3   | 60.9         | 80.5 | 91.4          | 21.3       |
> |√|×|90.7 |95.4 |	58.7	|79.3	|91.2 |	22.8 |
> |×|×|90.6|95.4|56.0|71.2|91.4|21.1|
>
> We conduct ablation study to prove the contribution of  block-averaging fusion module and the smoothed projection module. The average accuracies are reported. As the table shows, without two modules, the results of F-OAL are already competitive. The two  modules further improve F-OAL's performance, especially on fine-grained datasets (e.g., DTD, FGVCAircraft and Country211).
>
> ## Q3: Is there any work or experiment they can point to that suggests feature fusion works?
>
> The idea of feature fusion is inspired by DenseNet [1], which suggests that hidden features are still helpful for generating more representative final output. This approach can be considered equivalent to an ensemble of multiple backbones, which can provide feature diversity for F-OAL with a frozen encoder.
>
> [1] Densely Connected Convolutional Networks, CVPR 2017

---

> > ### Comment · Reviewer_nHHU · 2024-08-12
> > **Response to authors**
> >
> > Thank you for acknowledging and updating based on my remarks, these improvements are satisfactory in my view. However, I misunderstood the scoring procedure here. My score was contingent on those remarks being addressed, so it's unlikely to increase. I'm interested in the ongoing conversation with Reviewer UyXG as well, I hope that will be resolved before the closed discussion.

---

> > > ### Author Response · Authors · 2024-08-12
> > > **Thank you for the response**
> > >
> > > Thank you for taking the time to read our response! We are glad that the response addressed your concerns.

---

### Official Review · Reviewer_Pgch · 2024-07-21

**Soundness:** 2
**Presentation:** 2
**Contribution:** 2
**Rating:** 5
**Confidence:** 4

**Summary:**

The paper introduces Foward-only Online Analytic Learning (F-OAL), an exemplar-free approach designed for Online Class Incremental Learning. The method addresses Catastrophic Forgetting by utilizing a pre-trained frozen encoder and a recursive least square updated linear classifier, which significantly reduces memory usage and computational time. The authors conducted extensive experiments to demonstrate the effectiveness of F-OAL on multiple benchmark datasets, showing its superior performance over existing exemplar-free methods and several replay-based methods.

**Strengths:**

The F-OAL framework introduces a forward-only learning mechanism that avoids back-propagation, effectively reducing computational overhead and memory footprint.

By not relying on exemplar storage, F-OAL maintains data privacy, a crucial requirement in many real-world applications where data sensitivity is a concern.

**Weaknesses:**

While the paper is strong in many aspects, it lacks a detailed discussion on potential limitations of the proposed method, such as its dependence on the quality of the pre-trained encoder and the challenges that might arise in different data scenarios.

Although the paper compares several baseline methods, including more recent and varied techniques could provide a more comprehensive evaluation of F-OAL’s relative performance. Some recent exemplar-free works could be easily generalized to OCIL setting and should be considered include:

[1] Divide and not forget: Ensemble of selectively trained experts in continual learning, ICLR 2024

[2] R-dfcil: Relation-guided representation learning for data-free class incremental learning, ECCV 2022

[3] Self-sustaining representation expansion for non-exemplar class-incremental learning, CVPR 2022

[4] DiffClass: Diffusion-Based Class Incremental Learning, ECCV 2024

**Questions:**

Please refer to the weaknesses.

**Limitations:**

Please refer to the weaknesses. I think providing a more comprehensive comparison with recent state-of-the-art works, a complexity analysis and an overhead comparison would help justify the contribution of this work.

---

> ### Author Rebuttal · Authors · 2024-08-04
>
> # Replies to Reviewer Pgch
>
> Thank you for your constructive and detailed feedbacks. We provide detailed responses to your concerns below.
>
> ## W1: Dependence on the quality of the pre-trained encoder.
>
> Thank you for the suggestion. Indeed, our method relies on well pre-trained backbones such as ViT and ResNet pre-trained on ImageNet. We acknowledge this as a limitation and will discussion this limitation in the manuscript so readers can fully understand our technique.
>
> On the other hand, there are many open-source pre-trained backbones available within the deep learning community, which are relatively easy to obtain. Additionally, leveraging pre-trained models for fine-tuning downstream tasks has become a mainstream approach. Hence, although the need for pre-trained backbones is an existing limitation of F-OAL, it is feasible and in line with mainstream.
>
> ## W2:  Challenges that might arise in different data scenarios.
>
> Thank you for the suggestion. Indeed, we focus on coarse-grained data scenarios, such as CIFAR-100, Tiny ImageNet, and Core50, as well as fine-grained data scenarios, including DTD, FGVC Aircraft, and Country211. However, there are other data scenarios, such as **long-tail distributions**, which we have not addressed in this work. We shall include this discussion in the manuscript.
>
> ## W3: Including some recent exemplar-free works.
>
> Thank you for pointing out these baselines. However, they seem to be specially designed for non-pre-trained ResNet, and changing their backbones will compromise their performance. Therefore, we have included several alternatives, i.e., EASE [1], LAE [2] and SLCA [3], which are designed with pre-trained ViT and are the SOTA exemplar-free CIL approaches. **Please refer to the uploaded PDF in the general response.**
>
> In addition, the mentioned baselines [4-7] are good references, and we shall include them in our literature review to complete our review.
>
> [1] Expandable Subspace Ensemble for Pre-Trained Model-Based Class-Incremental Learning, CVPR 2024
>
> [2] A Unified Continual Learning Framework with General Parameter-Efficient Tuning, ICCV 2023
>
> [3] SLCA: Slow Learner with Classifier Alignment for Continual Learning on a Pre-trained Model, ICCV 2023
>
> [4] Divide and not forget: Ensemble of selectively trained experts in continual learning, ICLR 2024
>
> [5] R-dfcil: Relation-guided representation learning for data-free class incremental learning, ECCV 2022
>
> [6] Self-sustaining representation expansion for non-exemplar class-incremental learning, CVPR 2022
>
> [7] DiffClass: Diffusion-Based Class Incremental Learning, ECCV 2024
>
>
> ## L2: A complexity analysis.
>
> Thank you for the suggestion, we have included the complexity analysis as follows.
>
> In terms of space complexity, our trainable parameters are only  _**R**_  and _**W**_. The _**R**_ matrix is of size  D × D , where D is the output dimension of the encoder. In our paper, the encoder output dimension is 1000. Therefore, according to Equation 9, the size of the _**R**_ matrix is  1000 × 1000 . The _**W**_ matrix has dimensions of C × D , where C  is the number of classes in the target dataset. For example, with CIFAR100, its size is 100 × 1000 . The total number of trainable parameters is relatively small and does not require gradients. This results in our method using less than 2GB of memory, as shown in Figure 2.
>
> In terms of computational complexity, we denote the batch size as S, encoder’s output size as D, and class number as C. Therefore, the dimensions of $X$, $Y$, $R$ and $W$ are S×D, S×C, D×D and C×D, respectively. Thus, the calculation is shown below:
>
> The computational complexity for updating _**R**_ is dominated by the matrix multiplications, thus:
> $\text{max}${$\mathcal{O}(SDC), \mathcal{O}(SC), \mathcal{O}(SDC), \mathcal{O}(D^2C), \mathcal{O}(DC)$}≈ $\text{max}${$\mathcal{O}(SDC), \mathcal{O}(D^2C)$}
>
> The computational complexity for updating _**W**_ is dominated by the matrix multiplications and the matrix inversion:
>  $\text{max}${$\mathcal{O}(SD^2), \mathcal{O}(S^2), \mathcal{O}(S^3), \mathcal{O}(DS^2), \mathcal{O}(D^2S), \mathcal{O}(D^2)$} ≈ $\text{max}${$\mathcal{O}(S^3), \mathcal{O}(DS^2), \mathcal{O}(D^2S)$}
>
>
> In the OCIL setting, the batch size is relatively smaller. Therefore, the overall computational complexity is primally controlled by $D$.
>
> ## L3: Overhead comparison.
>
> In terms space overhead, compared to the conventional backbone + classifier structure, F-OAL introduces an additional linear projection to control the output dimension  D  of the encoder, and a matrix $R$ , where only $R$ is trainable. According to Equation 9, the dimension of  $R$  remains a fixed size of D × D. Other methods require more extra space. For instance, LwF employs knowledge distillation, necessitating the storage of additional models, while replay-based methods require extra storage to retain historical samples. In contrast, the overhead introduced by F-OAL, consisting of an additional matrix and a linear layer, is smaller.
>
> In terms of time overhead, our method primarily consists of a forward pass and matrix multiplication, which is determined by output dimension of encoder. By changing the output dimension of encoder, we can balance the accuracy and time.  According to [8], the backward pass in backpropagation (forward pass + backward pass) accounts for 70% of the time. Therefore, our method's time overhead is also relatively small.
>
> [8] Decoupled Parallel Backpropagation with Convergence Guarantee. ICML 2018
>
> Based on these additional results and clarifications, we hope you could consider increasing your score in support of this work. If not, could you kindly let us know what additionally needs to be done in your assessment to make this work ready for publication?

---

> > ### Comment · Reviewer_Pgch · 2024-08-13
> >
> > Thank you to the authors for their detailed response. I have also checked the feedback from other reviewers and will adjust my score accordingly.

---

> > > ### Author Response · Authors · 2024-08-14
> > > **Thank you for the response**
> > >
> > > Thank you for taking the time to read our respose. Could you kindly adjust our score if our response addresses your cocerns, before the discussion closes?  Thanks!

---

### Author Rebuttal · Authors · 2024-08-04

# General Response
We thank all the reviewers for their time, insightful suggestions and valuable comments. In summary, Reviewer nHHU appreciates that our work is **natural**, **intuitive**, and overall **quite strong**.  The writing is **clear**, **logical**, **informative** and **useful** for the reader. Reviewer UyXG points out that our empirical results are **very strong**. Both Reviewer Pgch and Reviewer XB6P appreciate that our work effectively reduces computational overhead and memory footprint.

We provide point-by-point responses to all reviewers’ comments and concerns. On the other hand, reviewers also point out the recursive mechanism is relatively rare in OCIL, and could not fully understand the F-OAL purely through the derivations. To address this in the response, we have attached the source code for a **quick experiment** comparing the BP and the F-OAL on MNIST dataset as follows. These codes should be able to run freely on any platform such as Colab.

Also the **PDF contains the table for Reviewer Pgch**.


```
import torch

import torch.nn as nn

import torch.optim as optim

from torchvision import datasets, transforms

import torch.nn.init as init

import torch.nn.functional as F


class MLP(nn.Module):

        def __init__(self):

            super(MLP, self).__init__()

            self.fc1 = nn.Linear(28 * 28, 1000)

            self.fc2 = nn.Linear(1000, 10, bias=False)

        def forward(self, x):

            x = self.get_activation(x)

            return self.fc2(x)

        def get_activation(self, x):

            x = x.view(-1, 28 * 28)

            x = torch.relu(self.fc1(x))

            return x

transform = transforms.Compose([transforms.ToTensor(), transforms.Normalize((0.1307,), (0.3081,))])

train_dataset = datasets.MNIST('./data', train=True, download=True, transform=transform)

test_dataset = datasets.MNIST('./data', train=False, download=True, transform=transform)

train_dataset_5 = [(data, target) for data, target in train_dataset if target < 5]

test_dataset_5 = [(data, target) for data, target in test_dataset if target < 5]

train_loader_5 = torch.utils.data.DataLoader(train_dataset_5, batch_size=64, shuffle=True)

test_loader_5 = torch.utils.data.DataLoader(test_dataset_5, batch_size=1000, shuffle=False)

model = MLP()

criterion = nn.CrossEntropyLoss()

optimizer = optim.SGD(model.parameters(), lr=0.01, momentum=0.9)


def train(model, train_loader, criterion, optimizer, epochs=1):

       model.train().cuda()

       for epoch in range(epochs):

           for data, target in train_loader:

               optimizer.zero_grad()

               output = model(data.cuda())

               loss = criterion(output, target.cuda())

               loss.backward()

               optimizer.step()


def test(model, test_loader):

        model.eval().cuda()

        correct = 0

        with torch.no_grad():

            for data, target in test_loader:

                output = model(data.cuda())

                pred = output.argmax(dim=1, keepdim=True)

                correct += pred.eq(target.cuda().view_as(pred)).sum().item()

        print(f'\nTest set: Accuracy: {correct}/{len(test_loader.dataset)} ({100. * correct / len(test_loader.dataset):.1f}%)\n')

        return 100. * correct / len(test_loader.dataset)


print('We train the MLP on first 5 classes of MNIST via 1 epoch of BP (OCIL setting)')

train(model, train_loader_5, criterion, optimizer)

print('\nThe test accuracy on the first 5 classes is:')

old_acc = test(model, test_loader_5)

train_dataset_5to9 = [(data, target - 5) for data, target in train_dataset if target >= 5]

test_dataset_5to9 = [(data, target - 5) for data, target in test_dataset if target >= 5]

train_loader_5to9 = torch.utils.data.DataLoader(train_dataset_5to9, batch_size=64, shuffle=True)

test_loader_5to9 = torch.utils.data.DataLoader(test_dataset_5to9, batch_size=1000, shuffle=False)

print('Then we train the MLP on the rest of 5 classes')

train(model, train_loader_5to9, criterion, optimizer)

print('We test the model on the old 5 classes to see if it forgets the old knowledge')

new_acc = test(model, test_loader_5)

print(f'The accuracy drops from {old_acc:.1f}% to {new_acc:.1f}%, suggesting that BP suffers a lot from CF')

print('\nNow we verify that our approach tackles CF')

print('For this easy dataset, we even do not need a powerful pre-trained backbone, and use the same MLP')

new_mlp = MLP()

R = (torch.eye(1000).float()).cuda().double()

W = (init.zeros_(new_mlp.fc2.weight.t())).double().cuda()


def trainFOAL(new_model, train_loader):

        global R, W

        new_model.train().cuda()

        with torch.no_grad():

            for data, target in train_loader:

                data, target = data.cuda(), target.cuda()

                activation = new_model.get_activation(data).double().cuda()

                label_onehot = F.one_hot(target, 10).double().cuda()

                R = R - R @ activation.t() @ torch.pinverse(
                    torch.eye(data.size(0)).cuda() +
                    activation @ R @ activation.t()) @ activation @ R

                W = W + R @ activation.t() @ (label_onehot - activation @ W)

                new_model.fc2.weight = torch.nn.parameter.Parameter(torch.t(W.float()))


print('Similarly, we still train our model on first 5 classes, and the test accuracy is:')

trainFOAL(new_mlp, train_loader_5)

old_FOAL_acc = test(new_mlp, test_loader_5)

print('Then we train our model on rest 5 classes, and the test accuracy on old classes is:')

trainFOAL(new_mlp, train_loader_5to9)

new_FOAL_acc = test(new_mlp, test_loader_5)

print(f'The small gap between {old_FOAL_acc:.1f}% and {new_FOAL_acc:.1f}% suggests that the our model does not forget the old knowledge')
```

---

### Decision · Program_Chairs · 2024-09-25

**Decision:**

Accept (poster)

**Comment:**

The reviewers were generally appreciative about the responses that authors gave to their concerns, and appreciated both the empirical results and computational efficiency of the proposed method. The authors are strongly encouraged to incorporate additional experiments and explanations they included in their rebuttal. These are, in particular:


* The extensive additional experiments against competitors included in the pdf of the rebuttal.
* The ablation w.r.t. capturing information from different levels of abstraction.
* The discussion on both computational and space complexity, as well as the comparison of overheads.
* A table with notations.
* A clear statement that evaluations include the feature extraction time.
* A discussion of limitations.
* A discussion on long-tailed distributions.
* A thorough correction of all typos indicated by reviewers, and a careful proofreading.